# The Shortening of Leukocyte Telomere Length Contributes to Alzheimer’s Disease: Further Evidence from Late-Onset Familial and Sporadic Cases

**DOI:** 10.3390/biology12101286

**Published:** 2023-09-26

**Authors:** Paolina Crocco, Francesco De Rango, Serena Dato, Rossella La Grotta, Raffaele Maletta, Amalia Cecilia Bruni, Giuseppe Passarino, Giuseppina Rose

**Affiliations:** 1Department of Biology, Ecology and Earth Sciences, University of Calabria, 87036 Rende, Italy; paolina.crocco@unical.it (P.C.); f.derango@unical.it (F.D.R.); serena.dato@unical.it (S.D.); rossella.lagrotta@unical.it (R.L.G.); giuseppe.passarino@unical.it (G.P.); 2Regional Neurogenetic Centre, ASP Catanzaro, 88046 Lamezia Terme, Italy; malettar@yahoo.it (R.M.); amaliaceciliabruni@gmail.com (A.C.B.)

**Keywords:** telomere length, telomeres, Alzheimer’s disease, late-onset AD, aging, neurodegeneration

## Abstract

**Simple Summary:**

Alzheimer’s disease (AD) is one of the most common forms of dementia in the aging population. The shortening of telomeres, which are complex structures at the ends of chromosomes, is considered one of the hallmarks of aging and has been implicated in several neurodegenerative diseases, especially AD, where the results are conflicting. Thus, to help clarify the association of telomere length with AD risk, leukocyte telomere length (LTL), measured as T/S ratio (telomere vs single-copy gene) was assessed in a cohort of 534 subjects, comprising sporadic and familial cases of late-onset AD (LOAD) and cognitively healthy controls. Compared with controls, LOAD cases showed significantly shorter telomeres. The association with disease risk was independent of confounders such as age, sex, Mini-Mental State Examination (MMSE), and Apolipoprotein E ε4 (APOE-ε4) status. Our findings support telomere shortening as a potential biomarker of LOAD risk.

**Abstract:**

Telomeres are structures at the ends of eukaryotic chromosomes that help maintain genomic stability. During aging, telomere length gradually shortens, producing short telomeres, which are markers of premature cellular senescence. This may contribute to age-related diseases, including Alzheimer’s disease (AD), and based on this, several studies have hypothesized that telomere shortening may characterize AD. Current research, however, has been inconclusive regarding the direction of the association between leukocyte telomere length (LTL) and disease risk. We assessed the association between LTL and AD in a retrospective case–control study of a sample of 255 unrelated patients with late-onset AD (LOAD), including 120 sporadic cases and 135 with positive family history for LOAD, and a group of 279 cognitively healthy unrelated controls, who were all from Calabria, a southern Italian region. Following regression analysis, telomeres were found significantly shorter in LOAD cases than in controls (48% and 41% decrease for sporadic and familial cases, respectively; *p* < 0.001 for both). Interestingly, LTL was associated with disease risk independently of the presence of conventional risk factors (e.g., age, sex, MMSE scores, and the presence of the APOE-ε4 allele). Altogether, our findings lend support to the notion that LTL shortening may be an indicator of the pathogenesis of LOAD.

## 1. Introduction

Alzheimer’s disease (AD) is a progressive and invalidating neurodegenerative disorder of aging. Decades of research have shed light on the disease’s neuropathological changes, which are characterized by neuronal loss and the aggregation of Aβ and tau proteins. Clinically, it is featured by gradual loss of memory, the impairment of cognitive ability, and the loss of independence, reflecting the underlying neurodegeneration. The predominant form of AD is late onset (LOAD; age of onset, over 65 years), which can be sporadic, the most common form accounting for 60–80% of dementia patients, or familial (15% to 20%). The pathophysiology of the disease is complex and, although several genomic loci and risk-enhancing genetic variants have been determined, the ε4 allele of the apolipoprotein E (APOE) gene and increasing age remain two of the most important known risk factors for the disease development. Affecting millions of people worldwide, LOAD represents one of the most common causes of dementia in the elderly and a serious public health problem [1]. So, an understanding of the underlying biopathological factors and biomarkers with prognostic significance is warranted.

Telomeres are complex nucleoprotein structures at the tip of each chromosome. They comprise repeating sequences of TTAGGG physically associated with the telomerase and a variety of telomeric proteins. Their function is to protect the ends of chromosomes from fusion and degradation events, thus preserving the genome stability of the cells. Several lines of evidence have been collected in recent years based on which the shortening of leukocyte telomere length (LTL) can be considered a critical cellular hallmark of biological aging. This view is supported by findings that shorter telomeres increase the risk of overall mortality and the risk of developing age-related disorders [2], such as cardiovascular disease, osteoporosis, type 2 diabetes mellitus, and cancers [3,4,5,6,7,8,9,10].

Several studies examined the association between LTL and LOAD, although with controversial results. While some case–control and meta-analysis studies found significantly shorter telomeres in individuals with LOAD than in healthy controls [11,12,13,14,15,16,17], others have not found such an association [18,19,20,21]. Also, a long-term longitudinal study by Fani and co-workers [22] reported a U-shaped association, suggesting both short LTL and long LTL as risk factors for AD. A nonlinear association was also found between LTL and mild cognitive impairment (MCI) [23]. In addition, it was reported that subjects with mild cognitive impairment that evolved into AD have longer LTL than those with stable mild cognitive impairment [21]. 

Overall, these results indicate that the association between LTL and LOAD is complex and requires additional investigation to untangle, also considering the importance that LTL may have as a biomarker with potential diagnostic and prognostic value in the assessment of patients with LOAD. Thus, in this study, we measured and compared the relative average LTL, determined by real-time quantitative PCR (RT-qPCR) and expressed as T/S ratio (telomere vs single-copy gene), among two groups of patients clinically diagnosed as LOAD, namely sporadic cases and patients who had a positive family history for LOAD, and a group of similarly aged cognitively healthy control subjects, to gain more insight into the relationship between LTL and LOAD risk.

## 2. Materials and Methods

### 2.1. Subjects

The sample analyzed was composed of 534 subjects from Calabria (southern Italy), comprising 255 patients with LOAD (95 men and 160 women), with a mean (±standard deviation, SD) age of 77.41 ± 2.80, and 279 unrelated healthy controls (147 men and 132 women), with a mean age of 73.67 ± 5.49. We paid special attention to the recruitment of the cases and controls to avoid false-positive results due to population stratification. To this aim, in this study, we only included subjects with at least three generations of ancestors from the Calabria region.

The control subjects were recruited during several campaigns focused on monitoring the quality of the elderly population in Calabria. The details of this sampling were previously reported [24]. Cases and controls were matched for age range, ethnicity, and geographical area. A detailed anamnesis and a rigorous clinical history were collected for all the subjects through a general and neurological examination, which included specific tests aimed to include/exclude the presence of any neurological disorder. A Mini-Mental State Examination (MMSE) test was administered for the assessment of cognitive status [25]. This test evaluates several different areas of thinking, including memory, judgment, calculation, abstraction, language, and visual–spatial ability, with scores ranging from 0 (lowest cognitive function) to 30 (highest cognitive function). The MMSE scores were adjusted for age and educational level according to the study of Magni and coworkers [26]. The patients were recruited from those admitted to the Regional Neurogenetic Center (ASP CZ), and the diagnosis of LOAD was performed according to the criteria of the National Institute on Aging and the Alzheimer’s Association workgroup [27]. McKeith criteria [28], clinical and neuropathological criteria for frontotemporal dementia [29], and NINDSAIREN criteria [30] were used for differentiating AD from Lewy body dementia, frontotemporal dementia, and vascular dementia.

The LOAD sample was further subdivided into sporadic (*n* = 120 subjects (45 men and 75 women) with a mean age of 76.9 ± 2.75) and familial (*n* = 135 subjects (50 men and 85 women) with a mean age of 77.86 ± 2.77) cases: If LOAD was diagnosed in one patient without further members of the family affected, the case was defined as “sporadic”. Conversely, if LOAD was diagnosed in a subject who had a positive family history for LOAD, the case was defined as “familial”. In such a case, one affected subject per family was randomly selected for the study. No known pathological mutations were detected in these patients.

### 2.2. Ethics Statement

This study was carried out in accordance with the ethical standards established in the Declaration of Helsinki and with the approval of the ethical committee (code n. 25/2017). Before the visit, all the patients signed an informed consent for the use of their clinical–pathological data and for permission to collect blood samples from which to extract genomic DNA for research purposes. When appropriate, a relative or legal representative signed the consent.

### 2.3. DNA Extraction

Genomic DNA was isolated from whole blood or buffy coats according to the salting-out procedure described by Miller et al. [31]. 

### 2.4. Leukocyte Telomere Length (LTL)

The average length of telomeres (LTL) was measured using a previously modified protocol described by Testa and colleagues [32], based on the quantitative real-time PCR-based assay. By using this technique, it is possible to quantify the relative telomere length as a T/S ratio evaluating the number of copies of telomeric repeats (T) in comparison to a single-copy gene (S) used as a quantitative control [33]. By following the modified methodology described by Testa [32], we used 36B4 as the single-copy gene, which encodes the acidic ribosomal phosphoprotein PO. For each sample, the number of copies of telomeric repeats (T) and the quantity of 36B4 reference gene (S) were determined in triplicate. The used primer sequences (5′→3′ direction) were tel 1, GGTTTTTGAGGGTGAGGGTGAGGGTGAGGGTGAGGGT; tel 2, TCCCGACTATCCCTATCCCTATCCCTATCCCTATCC-CTA; 36B4u, CAGCAAGTGGGAAGGTGTAATCC; and 36B4d, CCCATTCTATCATCAACGGGTACAA. To minimize inter-assay variability, the telomere, and single-copy gene (36B4) were assessed on the same plate. For the PCR reaction, two master mixes containing the PCR reagents, the SYBR green dye for the detection of the fluorescence, and the specific primers for telomeres (T) and the specific primers for 36B4 (S), were prepared using the recommended concentrations. Briefly, 15 µL of the mix was added in each well containing 15 ng of DNA (5 µL of DNA with a concentration of 3 ng/µL). For both telomere and 36B4, a calibrator DNA sample (Roche, Milan, Italy) (5 µL of 3 ng/µL) was added to each plate. Additionally, in each plate, the same reference DNA sample was used to prepare two standard curves (one for 36B4 and one for telomere reactions). In particular, the reference DNA sample (Roche, Milan, Italy) was diluted in series by 1.68-fold per dilution to produce six concentrations of DNA ranging from 30 to 2 ng in 5 µL. All PCRs were performed on the Applied Biosystems QuantStudio3 device in 96-well plates (Thermo Fisher Scientific, Waltham, MA, USA) using the thermal cycle profile consisting of (1) one cycle of 10 s at 95 °C and (2) 30 cycles of 5 s at 95 °C, 15 s at 57 °C, and 20 s at 72 °C. The samples were analyzed in triplicate, and the results are reported as T/S ratio relative to the calibrator sample to allow for comparison across runs. As a quality control, more than 20% of samples were randomly replicated on different plates. The correlation coefficient, r^2^, varied between 0.984 and 0.997, while the amplification efficiencies ranged from 94.8% to 107.5%, and the inter-assay coefficient of variation was lower than 7.8%.

### 2.5. APOE Genotyping

The two missense SNPs located in exon 4 of the *APOE* gene, rs429358-T/C at codon 112 (Arg/Cys), and rs7412-C/T at codon 158 (Arg/Cys), which determine the genotype of APOE for ε2, ε3, and ε4 protein isoforms, were genotyped using a polymerase chain reaction (PCR) amplification refractory mutation system (ARMS). For the analysis, the primers given by Wenham et al. [34] were used to identify the Arg/Cys polymorphism at codon 112 of the *APOE* gene (Arg 112 forward 5′-cgcggacatggaggacgttc-3′, Cys 112 forward 5′-cgcggacatggaggacgttt-3′, and common reverse primer 5′-gttcagtgattgtcgctgggca-3′), while to identify the Arg/Cys polymorphism at codon 158, primers given by Carrieri et al. [35] (Arg 158 forward 5′-atgccgatgacctgcagaggc-3′, Cys 158 forward 5′-atgccgatgacctgcagaggt-3′, and common reverse primer 5′-gtccggctgcccatctcctc-3′) were used. PCR was performed in a 20 μL reaction volume including 100 ng genomic DNA, 0.4 μL of Cys primers (10 μM) or Arg primers (10 μM), 0.8 μL of ARMS reverse primer (common primer; 10 μM), 1.6 μL of dimethylsulfoxide, 10 μL of green mix, and 4.8 μL of nuclease-free water. PCR amplification was initiated via the denaturation at 95 °C for 5 min, followed by amplification comprising 35 cycles of 95 °C for 30 s, 63 °C for 30 s, and 72 °C for 30 s, and final extension at 72 °C for 15 min. Amplified nucleotides were resolved via 2% agarose gel electrophoresis.

For the analysis, participants with APOE genotypes ε3/ε4 and ε4/ε4 were grouped as APOE ε4 carriers, while the rest of the genotypes were grouped as APOE ε4 noncarriers.

### 2.6. Statistical Analyses

Kolmogorov–Smirnov and Shapiro–Wilk tests were used to verify the normal distribution of the variables. Continuous variables are expressed as mean ± standard deviation (SD), while categorical variables are expressed as percentages. Differences between groups were evaluated using an independent sample *t*-test or chi-square test, for continuous and categorical values, respectively. Statistical comparisons among groups were performed by using one-way ANOVA followed by post hoc Tukey’s test. 

Telomere tertiles were formed based on the telomere length distribution in the whole sample after making age adjustments separately for case and control groups.

Linear regression analysis was carried out to assess any significant associations between telomere length and individual predictors, including age, age at onset, sex, MMSE score, and APOE status. The association between telomere length and LOAD was then assessed using forward stepwise multivariate logistic regression analyses while considering potential confounders. All statistical data were analyzed using the SPSS software version 28.0 (SPSS, Inc., Chicago, IL, USA). *p* < 0.05 was considered statistically significant.

## 3. Results

To investigate the contribution of leucocyte telomere length (LTL) to the risk of late-onset Alzheimer’s disease (LOAD), LTL, expressed as T/S ratio, was measured in a cohort of 255 unrelated patients with sporadic (sLOAD, 120 subjects) or familial (fLOAD, 135 patients) LOAD, and a similarly aged cohort of 279 cognitively normal controls. The characteristics of the samples are given in Table 1. Compared with controls, the affected individuals were older (mean age 76.9 ± 2.75 (sLOAD) and 77.86 ± 2.77 (fLOAD) vs. 73.67 ± 5.49 years) and with a lower proportion of males than the control group (37% vs. 52.7%). Moreover, in both case groups, the score of the risk variable MMSE was significantly lower than healthy controls, whereas the presence of the APOE e4 was significantly higher in the case groups than in controls. For all cases, *p* < 0.001 (Table 1).

For the analysis, LTL values were log-transformed to obtain a normal distribution.

As shown in Figure 1, the mean log-transformed LTL (logT/S ratio) in controls (−0.089, standard error (SE) 0.016) was significantly higher than that in sLOAD patients (−0.32, SE 0.017; *p* < 0.001), a difference that equates to about 48% decrease in the T/S ratio. Similarly, the mean logT/S ratio in controls was significantly higher than in fLOAD patients (−0.32, SE 0.019; *p* < 0.001), a difference that equates to a 41% decrease in the T/S ratio. No difference was observed in mean LTL between the two groups of patients (*p* > 0.05).

To better illustrate the telomere–LOAD relationship, the distribution of the logT/S ratio was categorized into tertiles constructed using the combined case–control group with the age adjustment carried out within the separate groups. The first tertile (ranging from −0.967 to −0.354, mean −0.46) represents the shortest telomeres, the second tertile (ranging from −0.353 to −0.137, mean −0.247) represents the medium, and the third tertile (ranging from—0.134 to 0.79, mean 0.112) represents the longest telomeres. Figure 2 shows the proportion of cases and controls across the three tertiles. As this figure shows, compared with controls, a significantly higher percentage of LOAD patients had telomeres in the first-tertile range, while, by contrast, a lower percentage of patients had telomeres in the third-tertile range (sex- and age-adjusted overall *p*-value < 0.001).

Linear regression analyses were performed, within each of the LOAD and control groups separately, to investigate the potential contribution of parameters reported in Table 1 to variation in LTL. We found a significant negative relationship between the logT/S ratio and age in the control group (β = −0.133, *p* = 0.027), while no significant relationship was observed between LTL and age in either subgroup of patients (β = −0.032, *p* = 0.73 for sLOAD; β = −0.014, *p* = 0.87 for fLOAD). No significant association with sex, age of disease onset, disease severity (as measured with MMSE score), or presence of the APOE-ε4 allele was found in any of the groups analyzed.

Next, we performed logistic regression analyses through a stepwise procedure to better evaluate the link between LTL and disease while including confounders in Table 1. The best model, reported in Table 2, included all the variables except sex. 

The results confirmed that shorter telomeres were significantly associated with increased risk of LOAD, both sporadic and familial LOAD (*p* < 0.001 for both), and that LTL was an independent risk factor for LOAD not confounded by the other risk indicators of the disease. Overall, these independent risk factors explained more than 70 percent (see Nagelkerke R2 values in Table 2) of the total variance in the predictive model performance.

## 4. Discussion

Telomere shortening is considered a marker of cellular aging. Yet, although the literature is extensive, evidence for an association with dementia, and more specifically with Alzheimer’s disease (AD) pathology, is more than controversial, with mixed results from different studies ranging from no association to negative or positive associations, as well as nonlinear associations [11,12,13,14,15,16,17,18,19,20,21,22,23]. These controversial results highlight the need for further investigation to gain more insight into the role that telomere length variability plays in LOAD risk. Toward this end, we analyzed a clinical-based series of sporadic and familial unrelated late-onset patients (sLOAD and fLOAD) and a cohort of cognitively normal subjects, all from a region of southern Italy (Calabria). Patients were older and with a higher proportion of females than the control group, probably reflecting the fact that the incidence of LOAD increases with age and is higher in women than men, as several studies have reported [36,37,38].

We found leukocyte telomere length (LTL) to be shorter in LOAD patients, both sLOAD and fLOAD cases, than in controls. This finding is supported by previous studies, among which is a well-powered meta-analysis including 13 studies, for a total of 860 patients with LOAD and 2022 controls [12]. Interestingly, short telomeres were also associated with reduced total and regional brain size [39]. These results and the findings of our study point to telomere biology as a potential pathway involved in the development of LOAD. However, although studies using Mendelian randomization provide support for a causal association between the risk of LOAD and short telomere length [40,41,42,43], establishing this causal link between shorter LTL and LOAD risk is particularly challenging. Many of the factors that induce accelerated telomere shortening, such as inflammation, oxidative stress, and immune function, have been implicated in LOAD. Therefore, it is possible that these factors, all likely embedded in a vicious cycle, where one feeds the other, may be part of the cause of telomere shortening and increased risk of LOAD and may potentially be at the basis of their relationship. The link between telomere shortening and LOAD has also, in part, been explained by connecting telomere shortening to the mechanisms controlling telomere maintenance. Wang and co-workers [44] have provided evidence that aggregated β-amyloid could inhibit telomerase activity, causing telomere shortening. In addition, Spilsbury et al. [45] reported that neurons expressing high levels of pathological tau did not express TERT (telomerase reverse transcriptase) protein, the expression of which has an impact on telomere length.

One additional finding of our study is that LTL is a significant independent risk factor for LOAD after multivariate adjustments for cognition-related confounders such as age, sex, MMSE score, and APOE status. The extant literature in this regard is inconclusive. Takata et al. [20] found that patients who are homozygous for APOE-ε4 have significantly shorter LTL than those with only one or no copies of APOE-ε4, a finding like that of Dhillon et al. [46], who reported telomeres significantly to be shorter in APOE-ε4 carriers than in non-APOE-ε4 carriers. On the other hand, Hackenhaar and co-workers [47] reported an association between short TL and a higher risk of AD in APOE non-ε4 carriers only, while Wikgren et al. [16] found that nondemented APOE-ε4 carriers had longer telomeres but a higher attrition rate than noncarriers. Differing from these studies, we did not detect any significant relationship between LTL and the presence of the APOE-ε4 allele in none of the analyzed cohorts. We also did not find a significant correlation between cognitive performance (MMSE scores) and LTL. This finding agrees with some but not all published studies, which report both shorter and longer telomeres associated with cognitive decline [48,49,50,51]. Inconsistency was also found between LTL and the rate of conversion to dementia in patients with mild cognitive impairment (MCI), a condition that can be a precursor to AD [51,52]. 

The lack of consistency across the results from different studies, regarding both the association between LTL with the disease and its relationship with other disease risk factors, could arise from different sources, including variability in methodologies used by different groups (study design, inclusion criteria, cell/tissue type examined, and measurement techniques), as well as differences in the ancestry of the populations studied. In terms of the latter factor, we want to point out that the three cohorts of subjects evaluated in the present study have some important homogeneity features. First, all subjects were collected in a population (Calabria, a region from southern Italy) characterized by a high level of genetic homogeneity due to the geographical and historical isolation of the region until recent years. Second, the samples were chosen among subjects with at least three generations of origin in the Calabria region. Therefore, population stratification, which is a significant confounding factor and a potential source of spurious associations, is limited in our sample. Third, a team of specialists like neurologists, neuropsychologists, and geriatricians made a huge effort to precisely define homogeneous phenotypes (for example, the distinction between sporadic and familial LOAD). Finally, the group of controls was matched with cases for ethnicity and genetic origin; moreover, and most importantly, the same neuropsychological tests used for cases were applied to the whole control sample to exclude the presence of latent forms of dementia. We should also point out that the same result was obtained for two independent cohorts of patients, namely sporadic cases and cases with a positive familial history of disease. We are therefore confident that the evidence emerging from our data, namely that shorter telomeres per se could be a risk factor for the development of LOAD, or nevertheless, a biomarker of the disease, is quite robust. 

We considered two alternative interpretations of the independent effect of LTL on LOAD: LTL may be associated with other risk factors for LOAD development not considered in this report; those persons with shorter telomeres may be more prone to develop the disease. In this regard, it is interesting to underline that we found a significant decrease in LTL with age in the control group but not in the LOAD groups, although the patients in the latter groups were older than the controls. This could indicate that individuals who develop LOAD inherently have shorter LTL that predisposes them to the disease; further physiological telomere shortening would lead to the death of cells with excessively short telomeres, thus reducing age-related LTL variability in these subjects. Alternatively, this could be associated with the fact that the rate of telomere attrition caused by the disease status, for instance, the LOAD-related increased oxidative stress and inflammation, which are among the main factors favoring telomere attrition, is significant enough in patients to mask the contribution of the gradual shortening of telomeres that normally occurs during aging.

### Study Limitations

The limitations of our study must be considered when assessing our findings. First, we measured telomere length in leukocytes in the blood, which may not be representative of telomere length in the brain, although some studies reported that telomere length in leukocytes is strongly correlated with that in the cerebellum of AD patients [53]. Second, its retrospective design, unlike the longitudinal design, does not allow for the determination of whether shorter telomeres in LOAD are a cause or rather a consequence of the disease; nor does it allow for the determination of whether there is a relationship between LTL and disease progression. Third, we did not collect additional information about the variables, biological or environmental (e.g., Aβ levels, unhealthy lifestyle factors, etc.), that could prove relevant to the results. 

## 5. Conclusions

In the present study, we measured the average length of leukocyte telomeres in two groups of LOAD patients, namely sporadic cases and cases with a positive familial history of disease, and cognitively healthy controls. All samples were from the same geographical area (Calabria, southern Italy). We found a significant decrease in LTL among sporadic and familial cases compared with controls (48% and 41% decrease, respectively). The association between shorter telomeres and the development of LOAD was independent of classical risk factors for LOAD like age, MMSE, sex, or the time of LOAD onset. Thus, our results further support the possibility that LTL may be important in the pathogenesis of this clinical entity, lending strength to the importance of telomere biology in disease development and the possibility of opening new avenues for further genetic investigation. However, no definite conclusions can be drawn as to this correlation, namely whether the observed shorter telomeres in patients are a cause or a consequence of disease, since this is a case–control study. Therefore, further investigations, possibly with a longitudinal design, are warranted to investigate the causal inferences about the association between telomere shortening and LOAD pathogenesis and to recognize shorter LTL as potential good predictive or diagnostic markers in the assessment of the disease.

## Figures and Tables

**Figure 1 biology-12-01286-f001:**
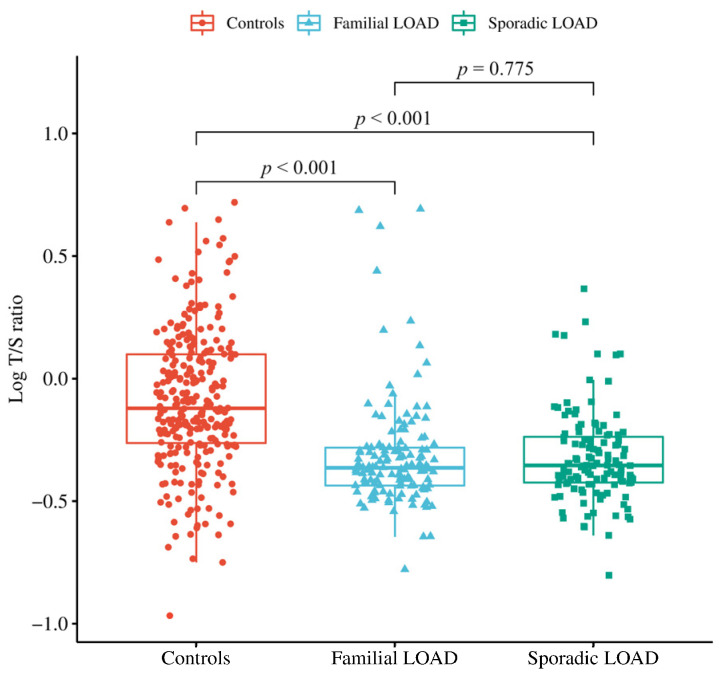
Mean values of peripheral blood leukocyte telomere length (LTL) measured as the logarithm of the number of copies of telomeric repeats (T) compared with a single-copy gene (S) (logT/S ratio) in sporadic and familial LOAD cases and controls. *p*-values were assessed using a one-way ANOVA adjusted with Tukey’s post hoc test.

**Figure 2 biology-12-01286-f002:**
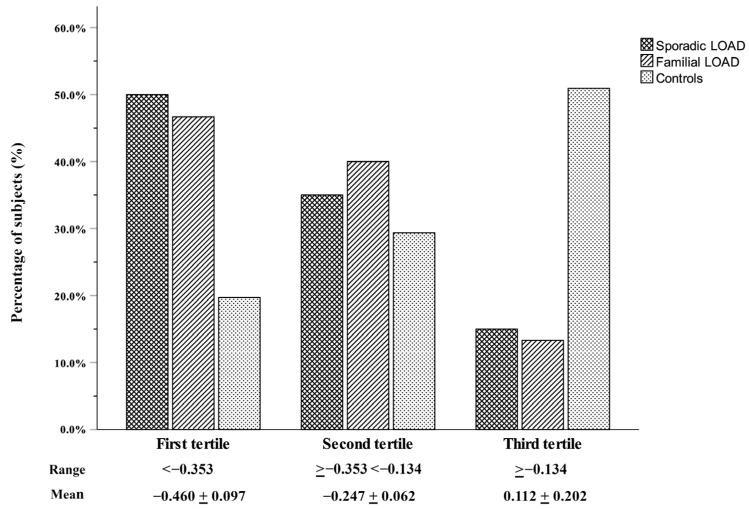
Stratification of the control group and patients with LOAD based on tertiles of age-adjusted log-transformed leukocyte telomere length, measured as the number of copies of telomeric repeats (T) in comparison to a single-copy gene (S) (logT/S ratio).

**Table 1 biology-12-01286-t001:** Baseline characteristics of the study participants.

	LOAD			Controls (*n* = 279)	
	Sporadic (*n* = 120)	Familial (*n* = 135)	All Patients (*n* = 255)		*p*-Value *
Age (mean ± SD)	76.9 ± 2.75	77.86 ± 2.77	77.41 ± 2.80	73.67 ± 5.49	<0.001
Males (%)	37.5	37.0	37.3	52.7	<0.001
Age onset (mean ± SD)	73.9 ± 5.40	73.72 ± 4.44	73.80 ± 5.41	-	
MMSE ** (mean ± SD)	14.34 ± 5.72	14.48 ± 5.01	14.42 ± 5.35	24.75 ± 3.73	<0.001
*APOE*-ε4 carriers (%)	38.3	43.7	41.2	8.6	<0.001

* *p*-value: total LOAD compared with the control; ** MMSE scores were adjusted for educational level and age at inclusion in the present study according to the procedure reported in Magni et al. [26].

**Table 2 biology-12-01286-t002:** Results of logistic regression analysis.

Variables	OR (95% CI) *	*p*-Value	Nagelkerke R2
	Sporadic LOAD		
MMSE scores	0.70 (0.65–0.76)	<0.001	0.712
LogTS ratio	0.03 (0.006–0.15)	<0.001	
APOE-ε4 status	5.89 (2.47–13.99)	<0.001	
Age	1.11 (1.03–1.19)	0.008	
	Familial LOAD		
MMSE scores	0.69 (0.64–0.75)	<0.001	0.734
LogTS ratio	0.09 (0.02–0.40)	<0.001	
APOE-ε4 status	4.33 (1.88–9.96)	<0.001	
Age	1.13 (1.04–1.22)	0.002	

* OR: odds ratio; CI: confidence interval.

## Data Availability

The data that support the findings of this study are available from the corresponding author upon request.

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
