# Peer review of "The Shortening of Leukocyte Telomere Length Contributes to Alzheimer’s Disease: Further Evidence from Late-Onset Familial and Sporadic Cases"

_biology, 2023, doi:10.3390/biology12101286_

Round 1

Reviewer 1 Report

The Article is devoted to the research of the role that telomere length plays in late-onset Alzheimer’s disease (LOAD) risk.

The specificity of the research is that the only subjects with at least three generations of ancestors from the Calabria region (Southern Italy) were included in the study. Cases and controls were matched for age range, ethnicity and geographical area.

An interesting and important problem is studied in the research, the obtained data should help to understand the late-onset Alzheimer’s disease mechanism.

The following comments do not diminish the value of the Article

Probably it would be better to specify in the aim as well as in the conclusions that the average leucocyte telomere length was measured.

Line 2 ‘Alz-‘ – probably the word in the title should not be transferred.

Line 23 Probably it would be better to decipher abbreviations (MMSE and APOE-ε4 status).

Line 84 Probably it would be better to specify the approach which’s been used to study telomere length.

Line 120 ‘2.2. Ethics statement.’ It is better to remove the dot at the end of the title.

Line 158 The following fragment ‘7.8 %.2.4.’ should be revised.

Line 194 The table and its title should appear on the same page ‘Table 1. Baseline characteristics of the study participants.’ It also would be good to add the information about the total amount of LOAD patients in the table.

According the data presented in the Table 1 age of LOAD patients differs from the age of Controls and it was higher proportion of females among affected individuals. Probably it would be good to discuss these data also, the possible reasons of these facts.

It would be better to include more details about the procedure of MMSE scores analysis (probably in Materials and Methods section), so the information presented in the Table 1 would be clearer.

It also would be better to describe a bit the specificity of APOE-ε4 carriers analysis, to understand better the data presented in the Table 1.

Line 198 It would be better to describe in a bit more details the specificity of data processing.

Lines 334-337 According to the presented statements probably it would be better to modify the title of the Article slightly.

References section should be checked and described according to the requirements published on the Journal website.

The scientific style has been used. A few sentences has to be revised.

Author Response

Responses to the reviewer comments on Ms. " The shortening of leukocyte telomere length contributes to Alzheimer's disease: further evidence from late‐onset familial and sporadic cases ".

We thank the reviewer for his/her thoughtful and thorough review and for the opportunity to revise and improve our Ms. We have gone through the comments and changed the Ms. according to them. We also provide a point-by-point answer to the comments. Changes in the main manuscript were reported in red.

Reviewer 1

The Article is devoted to the research of the role that telomere length plays in late-onset Alzheimer’s disease (LOAD) risk.

The specificity of the research is that the only subjects with at least three generations of ancestors from the Calabria region (Southern Italy) were included in the study. Cases and controls were matched for age range, ethnicity, and geographical area.

An interesting and important problem is studied in the research, the obtained data should help to understand the late-onset Alzheimer’s disease mechanism.

The following comments do not diminish the value of the Article.

  • Probably it would be better to specify in the aim as well as in the conclusions that the average leucocyte telomere length was measured.

We thank the reviewer for this suggestion. We modify the text accordingly.

  • Line 2 ‘Alz-‘ – probably the word in the title should not be transferred.

Thanks for the remark.

  • Line 23 Probably it would be better to decipher abbreviations (MMSE and APOE-ε4 status)

       Done

.

  • Line 84 Probably it would be better to specify the approach which’s been used to study telomere length.

 Done

  • Line 120 ‘2.2. Ethics statement.’ It is better to remove the dot at the end of the title.

Done

  • Line 158 The following fragment ‘7.8 %.2.4.’ should be revised.

Done

  • Line 194 The table and its title should appear on the same page ‘Table 1. Baseline characteristics of the study participants.’ It also would be good to add the information about the total amount of LOAD patients in the table.

Thanks for the suggestion. Accordingly,  we added in Table 1 information relative to the whole group of patients.

  • According the data presented in the Table 1 age of LOAD patients differs from the age of Controls and it was higher proportion of females among affected individuals. Probably it would be good to discuss these data also, the possible reasons of these facts.

Thank you. We added a sentence in the discussion section: “Patients were older and with a higher proportion of females with respect to the control group, probably reflecting the fact that the incidence of LOAD increases with age and is higher in women than men, as several studies report”

  • It would be better to include more details about the procedure of MMSE scores analysis (probably in Materials and Methods section), so the information presented in the Table 1 would be clearer.

           Thank you. We added the following sentence to the Materials and Methods section:

“This test evaluates several different areas of thinking including memory, judgment, calculation, abstraction, language, and visual–spatial ability, with scores ranging from 0 (lowest cognitive function) to 30 (highest cognitive function).”

  • It also would be better to describe a bit the specificity of APOE-ε4 carriers analysis, to understand better the data presented in the Table 1.

To clarify this point, we added this sentence to the Materials and Methods section :” For the .

       analysis, participants with the APOE genotypes ε2/ε4 ε3/ε4 and ε4/ε4 were grouped as APOE

      ε4 carriers, while the rest of genotypes were grouped as APOE ε4 non-carriers.)

  • Line 198 It would be better to describe in a bit more details the specificity of data processing.

Because the telomere length was skewed the values were log-transformed to obtain a normal distribution. We added this explanation to the results section in the new version of the Ms.

  • Lines 334-337 According to the presented statements probably it would be better to modify the title of the Article slightly.

We are sorry, but we did not find an agreement on a new title.

  • References section should be checked and described according to the requirements published on the Journal website.

Thanks for highlighting this. We carefully checked throughout the references.

Reviewer 2 Report

Dear esteemed authors, 

The paper exhibits a compelling and significant contribution to the scientific community's body of knowledge and research findings.

It would be advantageous to highlight the study's findings in the abstract section. 

The introduction serves to highlight the salient aspects and objectives of the investigation. It is suggested that there may be an opportunity for improvement in the structure of chapters within the introduction. 

The inclusion of the ethics committee statement procedure number is recommended in the "Materials and Methods" section. 

It is recommended to condense the results section. The document contains a substantial amount of information. 

The discussion chapter should prioritize the establishment of a parallel between your study and existing research on the subject matter, while also highlighting the potential innovative contributions of your study to the clinical and scientific domains of medicine and genetics. 

The inclusion of the study's limitations should be allocated to a distinct chapter, positioned between the sections dedicated to comments and conclusions.

The conclusions section might benefit from the inclusion of more material pertaining to the research. 

Author Response

Responses to the reviewer comments on Ms. " The shortening of leukocyte telomere length contributes to Alzheimer's disease: further evidence from late‐onset familial and sporadic cases ".

We thank the reviewer for his/her thoughtful and thorough review and for the opportunity to revise and improve our Ms. We have gone through the comments and changed the Ms. according to them. We also provide a point-by-point answer to the comments. Changes in the main manuscript were reported in red.

Reviewer 2

Dear esteemed authors,

The paper exhibits a compelling and significant contribution to the scientific community's body of knowledge and research findings.

  • It would be advantageous to highlight the study's findings in the abstract section.

Thank you for the suggestion. We tried to add some more details about findings in the abstract of the new Ms version.

  • The introduction serves to highlight the salient aspects and objectives of the investigation. It is suggested that there may be an opportunity for improvement in the structure of chapters within the introduction.

Following your suggestion, we re-structured the chapters within the introduction.  We hope this may improve the readability of the Ms.

  • The inclusion of the ethics committee statement procedure number is recommended in the "Materials and Methods" section.

Done

  • It is recommended to condense the results section. The document contains a substantial amount of information.

Thank you. We discussed about this point, but we do not consider proper to condense the result section because this could make difficult the understanding of the analysis especially to readers not particularly expert in the field.

  • The discussion chapter should prioritize the establishment of a parallel between your study and existing research on the subject matter, while also highlighting the potential innovative contributions of your study to the clinical and scientific domains of medicine and genetics.

In accordance with your comment, we added sentences in the Discussion section as follows:

We found leukocyte telomere length (LTL) shorter in LOAD patients, both sLOAD and fLOAD cases, than in controls. This finding is supported by previous research among which a well-powered meta-analysis including 13 studies for a total 860 patients with LOAD and 2022 controls [12]. Interestingly, short telomeres were also associated with reduced total and regional brain size [39]. These and our study point to telomere biology as a potential pathway involved in the development of LOAD. However, although studies using Mendelian randomization provide support for a causal association between risk of LOAD and short telomere length [40-43], to establish this causal link between shorter LTL and LOAD risk is particularly challenging.” and  in conclusion: …“Thus, our results further support the possibility that LTL may be important in the pathogenesis of this clinical entity, lending strength to the importance of telomere biology in disease development and the possibility of opening new avenues for further genetic investigation.”

  • The inclusion of the study's limitations should be allocated to a distinct chapter, positioned between the sections dedicated to comments and conclusions.

          As suggested, we allocated the limitations in a distinct paragraph.

  • The conclusions section might benefit from the inclusion of more material pertaining to the research.

Thanks for your suggestion. Accordingly, we included in the conclusions section more information.

Reviewer 3 Report

In the manuscript entitled “The shortening of leukocyte telomere length contributes to Alzheimer's disease: further evidence from late‐onset familial and sporadic cases” authors descrive the results obtained from a study in which the Leukocyte Telomerase Length (LTL) has been studied to propose it as a tentative biomarker in the diagnosis and prognosis of late onset Alzheimer disease (LOAD) subjects. To perform this study, authors have enrolled 255 subjects with LOAD, sporadic and familiar LOAD, and compared with an age-matched healthy control group made of 279 subjects.

The rational for conducting this study arises from the research performed by Lukens and collaborators [1] (Lukens et al., 2009) who described a significant correlation between peripheral blood leukocyte (PBL) telomere lengths and cerebellum telomere lengths in AD patients, though they did not find differences between AD patients cerebellum telomere lengths and age-matched control subjects. In this study, a very low number of subjects were enrolled (22 healthy control subjects and 30 AD patients).

 Overall, the manuscript is written in a comprehensible and scientific style and is well presented. Nevertheless there are some items that should be corrected/improved. They are listed bellow:

-          Pg 1, lines 15-18, Simple summary, there is a repeated meaning sentence:

“Alzheimer’s disease (AD) is one of the most common forms of dementia in the aging population, and among the hallmarks of aging, the shortening of telomere, which are complex structures at the ends of chromosomes, has been shown to contribute to aging in both animals and humans”.

“Telomere shortening has also been seen in several neurodegenerative diseases, especially AD where results are conflicting”. These two sentences should be rewritten in one sentence.

-          Simple summary, lines 19-20, it is not clear what “a proxy of telomere length” means when explaining  T/S ratio. It should be clarified but in the methods part of the manuscript, better than in the simple summary.

-          Abbreviations should be described the first time they appear in the text. In the manuscript some abbreviations have not been described (e.g., MMSE, SE, OR, TERT…  ).

-          When describing the composition and age of the different experimental groups, authors should better explain the related information they are giving. For instance, at pg 2, lines 90-91, “…255 patients with LOAD (95 men and 160 women; mean ages 77.41±2.80) and 279 unrelated healthy controls (147 men and 132 women; mean ages…” what theses numbers refeer to? is 77.41±2.80 the MEAN ± SEM, SD???

-          Correct “mean age”, not “mean ages” through all the manuscript.

-          Pg 3, lines 122-123, the sentence “…. All the patients or, where appropriate, a relative or legal representative, signed, before the visit,…” is not clear. Do authors mean that “All the patients signed an informed consent for the use of their clinico pathological data before the visit. When appropriate, a relative or legal representative signed the consent”.

-          The method to extract the genomic DNA has not been described, it should be included in the manuscript.

-          Pg 4, lines 152-156, the text should be better described. “The measurements” should be changed with “samples”. “Samples were performed in triplicate and results were reported as T/S…”

Does the blindly influence the reproducibility of experiment resuls?

This part of the paragraph is not clear.

-          The number “2.4” corresponding to the APOE genotyping method description is nota t the right line, 158 instead of 159.

-          Pg 4, lines 164 and 166, the description of primers used and the real time PCR protocol should be included in the text of the manuscript.

-          Pg 4, Results section, there is a mistake. Do authors mean “lower proportion of males with respect to the control group (37% vs 52.7%)”.

-          Describe the data included in Table 1, are they mean ± SD or SE?

-          The exact value of “P” obtained from statistical analysis should be specified through all the manuscript.

-          Figure 2, the title of the Y-axis is missing.

-          Pg 6, lines 215-217, it is not clear whe result values are positive or negative. Better write:

“…(range -0.967 to -0.354, mean 0.46)…”,

“… (range -0.353 to -0.137, mean 0.247)…”

“…(range -0.134 to -0.79, mean 0.112)…”.

-          Pg 9, lines 287-291, too long sentence, the meaning is not clear. Rewrite it.

-          Pg 9, line 294, “arise” instead of “arises”.

Pg 9, lines 318-321, again too long sentence, the meaning is not clear. Rewrite it. 

Author Response

Responses to the reviewer comments on Ms. " The shortening of leukocyte telomere length contributes to Alzheimer's disease: further evidence from late‐onset familial and sporadic cases ".

We thank the reviewer for his/her thoughtful and thorough review and for the opportunity to revise and improve our Ms. We have gone through the comments and changed the Ms. according to them. We also provide a point-by-point answer to the comments. Changes in the main manuscript were reported in red.

Reviewer 3

Comments and Suggestions for Authors

In the manuscript entitled “The shortening of leukocyte telomere length contributes to Alzheimer's disease: further evidence from late‐onset familial and sporadic cases” authors descrive the results obtained from a study in which the Leukocyte Telomerase Length (LTL) has been studied to propose it as a tentative biomarker in the diagnosis and prognosis of late onset Alzheimer disease (LOAD) subjects. To perform this study, authors have enrolled 255 subjects with LOAD, sporadic and familiar LOAD, and compared with an age-matched healthy control group made of 279 subjects.

The rational for conducting this study arises from the research performed by Lukens and collaborators [1] (Lukens et al., 2009) who described a significant correlation between peripheral blood leukocyte (PBL) telomere lengths and cerebellum telomere lengths in AD patients, though they did not find differences between AD patients cerebellum telomere lengths and age-matched control subjects. In this study, a very low number of subjects were enrolled (22 healthy control subjects and 30 AD patients).

Overall, the manuscript is written in a comprehensible and scientific style and is well presented. Nevertheless there are some items that should be corrected/improved. They are listed bellow:

  • Pg 1, lines 15-18, Simple summary, there is a repeated meaning sentence:

“Alzheimer’s disease (AD) is one of the most common forms of dementia in the aging population, and among the hallmarks of aging, the shortening of telomere, which are complex structures at the ends of chromosomes, has been shown to contribute to aging in both animals and humans”.

“Telomere shortening has also been seen in several neurodegenerative diseases, especially AD where results are conflicting”. These two sentences should be rewritten in one sentence.

Thanks. as suggested the two sentences were rewritten in one sentence.

  • Simple summary, lines 19-20, it is not clear what “a proxy of telomere length” means when explaining T/S ratio. It should be clarified but in the methods part of the manuscript, better than in the simple summary.

Thanks, you are right that “a proxy of telomere length” may be misleading. Thus, we deleted this phrase.

  • Abbreviations should be described the first time they appear in the text. In the manuscript some abbreviations have not been described (e.g., MMSE, SE, OR, TERT… ).

Done

  • When describing the composition and age of the different experimental groups, authors should better explain the related information they are giving. For instance, at pg 2, lines 90-91, “…255 patients with LOAD (95 men and 160 women; mean ages 77.41±2.80) and 279 unrelated healthy controls (147 men and 132 women; mean ages…” what theses numbers refeer to? is 77.41±2.80 the MEAN ± SEM, SD???

Thank you for pointing this out. We corrected making the sentence clearer. We specified that age was expressed as mean ± standard deviation.

  • Correct “mean age”, not “mean ages” through all the manuscript.

Done

  • Pg 3, lines 122-123, the sentence “…. All the patients or, where appropriate, a relative or legal representative, signed, before the visit,…” is not clear. Do authors mean that “All the patients signed an informed consent for the use of their clinico pathological data before the visit. When appropriate, a relative or legal representative signed the consent”.

Thank you, we changed as you suggested.

  • The method to extract the genomic DNA has not been described, it should be included in the manuscript.

Done

  • Pg 4, lines 152-156, the text should be better described. “The measurements” should be changed with “samples”. “Samples were performed in triplicate and results were reported as T/S…”

Thanks. We changed the sentence as you suggested.

  • Does the blindly influence the reproducibility of experiment resuls? This part of the paragraph is not clear.

The reviewer is right, the sentence in not very clear. We changed the sentence as follow: “As      a quality control, more than 20 % of samples were randomly replicated on different plates.

  • The number “2.4” corresponding to the APOE genotyping method description is nota t the right line, 158 instead of 159.

Done

  • Pg 4, lines 164 and 166, the description of primers used and the real time PCR protocol should be included in the text of the manuscript.

As required, we added the primers description. As for the protocol, we did not used real time PCR but ARMS. We added the relative protocol in the new version of Ms.

  • Pg 4, Results section, there is a mistake. Do authors mean “lower proportion of males with respect to the control group (37% vs 52.7%)”.

Thank you for highlighting this mistake. We made correction

  • Describe the data included in Table 1, are they mean ± SD or SE?

Data in table 1 are describes a mean ± SD, while data referring to Figure 1 are reported as mean ± SE

  • The exact value of “P” obtained from statistical analysis should be specified through all the manuscript.

We are sorry, but the program of analysis we used does not return an exact p-value when this is less than 0.001. In fact, when the value is higher than this limit, we reported the exact value.

  • Figure 2, the title of the Y-axis is missing.

Done

  • Pg 6, lines 215-217, it is not clear whe result values are positive or negative. Better write:

“… (range -0.967 to -0.354, mean 0.46)…”,

“… (range -0.353 to -0.137, mean 0.247)…”

“…(range -0.134 to -0.79, mean 0.112)…”.

Thanks. Done

  • Pg 9, lines 287-291, too long sentence, the meaning is not clear. Rewrite it

Thanks. We rewrite the sentence as follows: “We also did not find significant correlation between cognitive performance (MMSE scores) and LTL. This finding agrees with some but not all published studies which re-port both shorter and longer telomeres associated with cognitive decline [48-51]. Inconsistency was also found between LTL and rate of conversion to dementia in patients with mild cognitive impairment (MCI), a condition that can be a precursor to AD [51,52].”

  • Pg 9, line 294, “arise” instead of “arises”.

Done

  • Pg 9, lines 318-321, again too long sentence, the meaning is not clear. Rewrite it.

Thanks. We rewrite the sentence as follows: “It is interesting to underline on this regard that we found a significant decrease of LTL with age in the control group but not in LOAD groups, although these last were older than controls.”
